# An Expanded Investigation of Atmospheric Rivers in the Southern Appalachian Mountains and Their Connection to Landslides

**Douglas K. Miller** [1,*] , **Chelcy Ford Miniat** [2] , **Richard M. Wooten** [3] **and Ana P. Barros** [4]

1   Atmospheric Sciences Department, University of North Carolina Asheville, Asheville, NC 28804, USA
2   Coweeta Hydrologic Laboratory, United States Department of Agriculture, Otto, NC 28763, USA; cfminiat@fs.fed.us
3   North Carolina Geological Survey, Swannanoa, NC 28778, USA; rick.wooten@ncdenr.gov
4   Civil and Environmental Engineering, Duke University, Durham, NC 27708, USA; barros@duke.edu
*   Correspondence: dmiller@unca.edu; Tel.: +1-828-232-5158

**Abstract:** Previous examination of rain gauge observations over a five-year period at high elevations within a river basin of the southern Appalachian Mountains showed that half of the extreme (upper 2.5%) rainfall events were associated with an atmospheric river (AR). Of these extreme events having an AR association, over 73% were linked to a societal hazard at downstream locations in eastern Tennessee and western North Carolina. Our analysis in this study was expanded to investigate AR effects in the southern Appalachian Mountains on two river basins, located 60 km apart, and examine their influence on extreme rainfall, periods of elevated precipitation and landslide events over two time periods, the 'recent' and 'distant' past. Results showed that slightly more than half of the extreme rainfall events were directly attributable to an AR in both river basins. However, there was disagreement on individual ARs influencing extreme rainfall events in each basin, seemingly a reflection of its proximity to the Blue Ridge Escarpment and the localized terrain lining the river basin boundary. Days having at least one landslide occurring in western North Carolina were found to be correlated with long periods of elevated precipitation, which often also corresponded to the influence of ARs and extreme rainfall events.

**Keywords:** atmospheric rivers; extreme rainfall; landslides; southern Appalachian Mountains; Maya Corridor

## 1. Introduction

Atmospheric Rivers (ARs) are narrow and elongated zones of rapid, anomalously moist air at low levels originating from the sub-tropics and located just ahead of the surface cold front in mid-latitude cyclones, responsible for a significant portion of poleward vapor and latent heat transport (e.g., References [1–4]). These important features can be responsible for just over half of the extreme (top 2.5%) rainfall events in the Southeastern U.S. and societal hazards originate from a majority of these events (e.g., flooding, landslides) [5]. Just as in the eastern U.S., ARs in the U.S. West Coast of North America are responsible for extreme rainfall and flooding events, and subsequent landslides, but are also responsible for a high percentage of annual precipitation in what is typically an arid region (e.g., References [6–12]). Landslides in the Western U.S. are also more likely to occur in conjunction with ARs on lands affected by wildfires [12]. The importance of ARs has been documented in a variety of climate zones, and across the globe (e.g., References [13–28]).

Precipitation events in the southeastern U.S. have an abundance of sources from which to draw available water vapor, with the warm surface waters of the western Atlantic Ocean, Gulf of Mexico,

and Caribbean Sea located in close proximity. Not all extreme rainfall events in the Southeast are directly linked to an AR, and not every AR is directly responsible for an extreme rainfall event [5,22,29]. Although the same generally applies to the western U.S., the linkage there among ARs, extreme rainfall events, and dangerous societal impacts is more direct due to its arid climate and generally low background water vapor amounts.

Given the indirect nature of linkage among ARs, extreme rainfall and landslides in the Southeast, understanding the factors that make an AR-influenced event more likely to bring extreme rainfall and dangerous societal impacts to this region is imperative for forecasting. Recent studies show that both meteorological and physiographic conditions may help predict whether extreme rainfall results from ARs. For example, a slowly propagating 'hang-back' positively-tilted trough described in Miller et al. [5] allowed a single strong and intense slow-moving mid-latitude storm, and associated AR, to develop and deposit its heavy precipitation over the southern Appalachian Mountains. Locations that were in the rain-shadow of the local prominent orography only showed modest precipitation from such AR-influenced events [5]. Remnants of landfalling hurricanes and their associated extreme rainfall generally are responsible for a large number of landslides occurring within a relatively short period in the southern Appalachians [30]. Antecedent moisture conditions and soil type are critical for determining if an AR or landfalling tropical cyclone produces flooding and positive pore pressures that can trigger landslides. Whether these meteorological and physiographic predisposing conditions are general ones that may be applied to the larger region is unknown, however.

Landslides are a common occurrence in the southern Appalachian Mountains [30,31]. Most examples of severe landslide events in the region where hundreds to thousands of landslides, and debris flows, in particular, have occurred are associated with rainfall from remnant hurricanes (dying tropical cyclones). For example, the remnants of Hurricane Camille in 1969 generated over 5000 landslides in the mountains of Virginia and West Virginia; the remnants of Hurricanes Frances and Ivan in 2004 were responsible for at least 400 landslides in western North Carolina [32]. Although not generally linked to as many historically-disastrous events, mid-latitude storms, either individually, or as back-to-back events, have also been responsible for events in which hundreds of landslides have been generated [30].

Here we investigate the difference in response of two river basins located within 60 km of each other in the southern Appalachian Mountains, Pigeon River Basin (PRB) and Coweeta River Basin (CRB), to the passage of atmospheric rivers in the region. We also examine a long-term precipitation record of the CRB (initiated in 1934) to investigate the frequency that ARs play either in the role of pre-conditioner (increasing the soil moisture) or initiator (heavy rainfall) of landslides in the region [30]. This latter objective required developing a long-term data series of ARs that pre-dated current observation systems for the region. Dated technology of observing systems limits the accuracy of remotely-sensed information utilized by data assimilation techniques that allow continuous reanalysis of past weather events. These techniques can lead to degraded analyses as events are examined farther in the past. This is a particular problem when utilizing gridded moisture analyses in the vicinity of mountain ranges. Hence, we developed a new method of detecting ARs for investigating case studies in the distant past, and compare them with current methodology for detecting ARs over an eight-year study period. We analyze daily precipitation observations from the PRB and CRB, upper-air sounding observations, landslides documented by the North Carolina Geological Survey (NCGS), Global Forecast System (GFS) analyses of the more recent weather events, and Twentieth Century Reanalysis Project (Version 2c, [33]) case study analyses of events dating back to the mid-1900s and summarize findings and their implications for making improved heavy precipitation accumulation forecasts and debris flow predictions for the southern Appalachian Mountains.

## 2. Materials and Methods

The foundation of this study is built upon rainfall observations of two unique networks located in the southern Appalachian Mountains; the Duke Great Smoky Mountains Rain Gauge Network (hereafter referred to as 'Duke GSMRGN', Figure 1, Table A1; [5,34]) located in the PRB and the

climate and precipitation stations of the Coweeta Hydrologic Laboratory (hereafter referred to as 'CHLRGN', Figure 1, Table A1; [35]) located in the CRB. The former network was installed, starting in 2007, partly to improve the understanding of precipitation production mechanisms in the southern Appalachian Mountains after severe flooding occurred in September 2004 associated with the remnants of Hurricanes Frances and Ivan. It was also installed to aid in ground validation efforts of NASA's Global Precipitation Measurement Mission core satellite launched in February 2014 [34,36]. The latter network was installed soon after the establishment of the USDA Forest Service Coweeta Hydrologic Laboratory in 1934 [37]. We use data from both networks to examine the 'recent' past (1 July 2009–30 June 2017) and use data from the CHLRGN to examine the 'distant' past (4 June 1936–30 June 2017).

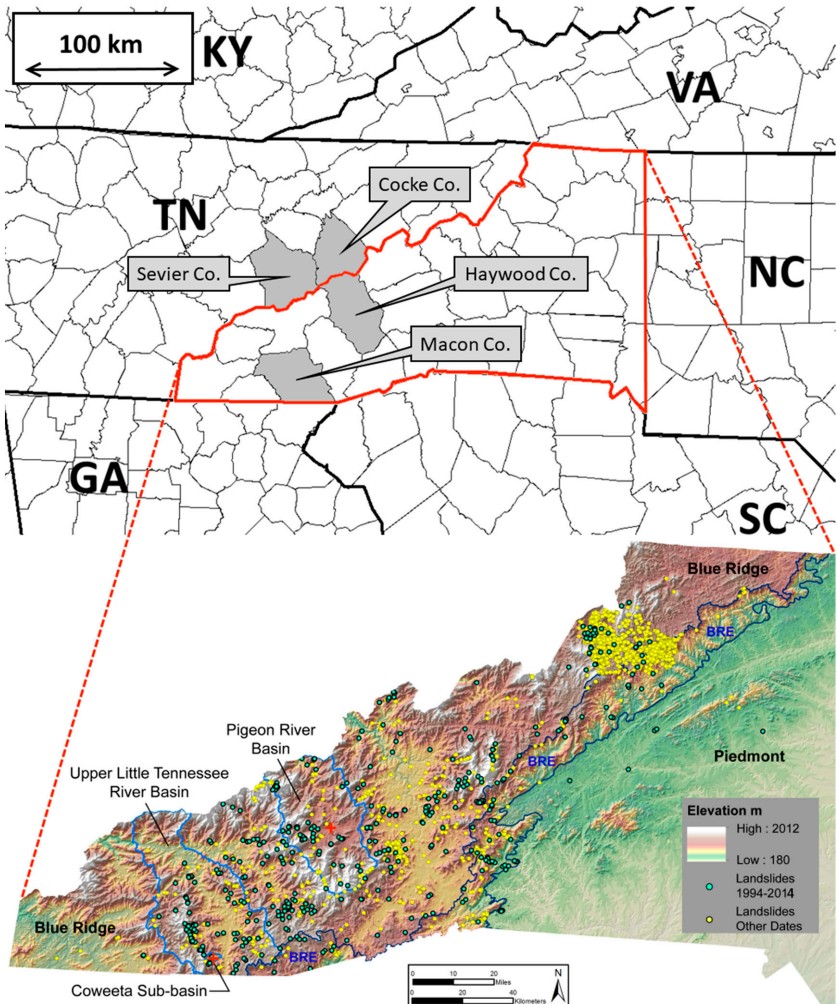

**Figure 1.** Locations of the Pigeon River Basin and Coweeta River Basin, a sub-basin of the Upper Little Tennessee Basin, and topographic elevation of the southern Appalachian Mountains. The Pigeon River Basin (PRB) corresponds to the borders of Haywood County, North Carolina and extends northward slightly into Cocke and Sevier Counties, Tennessee. The Coweeta River Basin (CRB) is located in Macon County, North Carolina. Specifics on the locations and elevations of individual rain gauges of the Duke GSMRGN, located in the North Carolina region of the PRB, and CHLRGN, located in the CRB, are provided in Table A1. Positions of the center points of each basin relative to each other and to the Blue Ridge Escarpment (labeled 'BRE' and outlined) are indicated with a red '+' located 60 km apart. The Blue Ridge Escarpment is the boundary between the Blue Ridge and the Piedmont physiographic province. Green dots highlight 529 landslide locations documented by the NCGS that initiated during the 21-year period, 1994–2014. The map base is a shaded relief map and color-coded elevation map derived from a 20 m pixel resolution LiDAR (Light Detecting and Ranging) digital elevation model.

Landslide inventory data for North Carolina used in the study came from the landslide geodatabase maintained by the NCGS [30]. The geodatabase documents 529 landslides of various types for the 1994–2014 focus period of this study (green dots in Figure 1), where the known date(s) of movement for individual landslides are recorded in the geodatabase.

## 2.1. Examination of ARs in the 'Recent' Past (1 July 2009–30 June 2017)

Sub-daily total rainfall accumulation observations over an eight year period from 32 tipping-bucket gauges of the Duke GSMRGN (Figure 1, Table A1) and nine recording rain gauges at CHLRGN (Figure 1, Table A1; [38]) were used to examine differences in the response of rainfall in the PRB (area of 1823 km$^2$) and the CRB (area of 16.3 km$^2$) to AR-influenced weather events, respectively. The latter is a sub-basin of the Upper Little Tennessee River Basin, located in close proximity to the Blue Ridge Escarpment (Figure 1). The Duke GSMRGN gauges are located at relatively high elevations, ranging from 1036 to 2003 m Above Sea Level (ASL, Table A1), while the CHLRGN gauges range 687 to 1366 m in ASL elevation (Table A1). The Duke GSMRGN network covers most of Haywood County, North Carolina (NC) and a portion of Cocke and Sevier Counties in eastern Tennessee (Figure 1). The CHLRGN covers a portion of Macon County, NC and is located close to the North Carolina and South Carolina border. The center point of each basin lies approximately 60 km apart, along a line oriented southwest and northeast (Figure 1).

Following the methodology of Miller et al. [5], total rainfall accumulation observed by the two rain gauge networks was binned into synoptic 6-h periods (0000–0600 Coordinated Universal Time (UTC), 0600–1200 UTC, 1200–1800 UTC, and 1800–0000 UTC) corresponding to the 6-h time resolution of the Global Forecast System (GFS) analysis National Centers for Environmental Information (NCEI) NOMADS archives. Total amounts recorded at or after the start of the 6-h period and before the end of the period were summed into a single bin. Events were defined as having concluded when no amounts were recorded at any of the network gauges during at least a single synoptic 6-h period [39]. Rainfall accumulation at each gauge during each 6-h period was summed and the total divided by the number of reporting gauges, giving the per gauge accumulation. Non-zero per gauge accumulation amounts of each consecutive synoptic 6-h period were added to calculate the event total per gauge accumulation. Events at each gauge network were defined separately to capture the influence on precipitation production of local orography.

The method of detecting ARs during the eight years study period also follows Miller et al. [5], which builds on the techniques of Wick [40] and Mahoney et al. [22]. GFS gridded analyses were used to generate vertically-integrated horizontal water vapor transport (IVT; [9]) fields over the 1000–100 hPa layer for every available 6-h GFS analysis time period available in the NCEI NOMADS archive (11,463 of 11,688 possible 6-h periods were located, covering 98.1% of the eight years study period). Periods missing from the GFS analysis archive were removed from consideration in this study. IVT was calculated as

$$-\int_{po}^{p}(q\mathbf{V})\frac{dp}{g},\qquad(1)$$

where $q$ is the specific humidity, $\mathbf{V}$ is the horizontal wind, $po$ is 1000 hPa, $p$ is 100 hPa, and $g$ is the acceleration due to gravity. GFS-based analysis IVT fields were examined closely over the IVT study domain, a 10° longitude by 5° latitude region centered in longitude on 83° W, and directly south of, the center of the PRB (black boundary in Figure 2). The IVT threshold and shape requirements of Mahoney et al. [22], 500 kg m$^{-1}$ s$^{-1}$, with the IVT feature no greater than 1500 km in width and no less than 1500 km in length (elliptical rather than circular shape), were used to detect ARs in the IVT study domain objectively. Our modified AR detection algorithm searched for IVT features located within the IVT study domain (Figure 2) and required that the feature influenced the domain for at least 12 h (3 consecutive GFS analysis times) with a mean IVT of at least 500 kg m$^{-1}$ s$^{-1}$ over the 12-h period. A minimum duration of eight hours was required in the studies of Neiman et al. [41,42] and

Ralph et al. [43] along the U.S. west coast. The more conservative time restriction of 12 h was selected because archives of GFS-analysis fields were only available every six hours.

The method of detecting ARs in the southeastern U.S. is more complicated than in the western U.S. due to the potential contribution to large horizontal water vapor transport by frontal systems, remnants of tropical cyclones, and mesoscale convective systems. The shape and minimal time requirements and relatively coarse horizontal grid spacing of the gridded GFS analyses ($0.5°$ latitude $\times$ $0.5°$ longitude) used by the Miller et al. [5] detection algorithm effectively eliminated mesoscale convective systems and frontal systems from consideration. The best track data (HURDAT2) of the National Hurricane Center [44] was used to flag landfalling hurricanes located near the IVT study domain and excluded them from the AR database of this study.

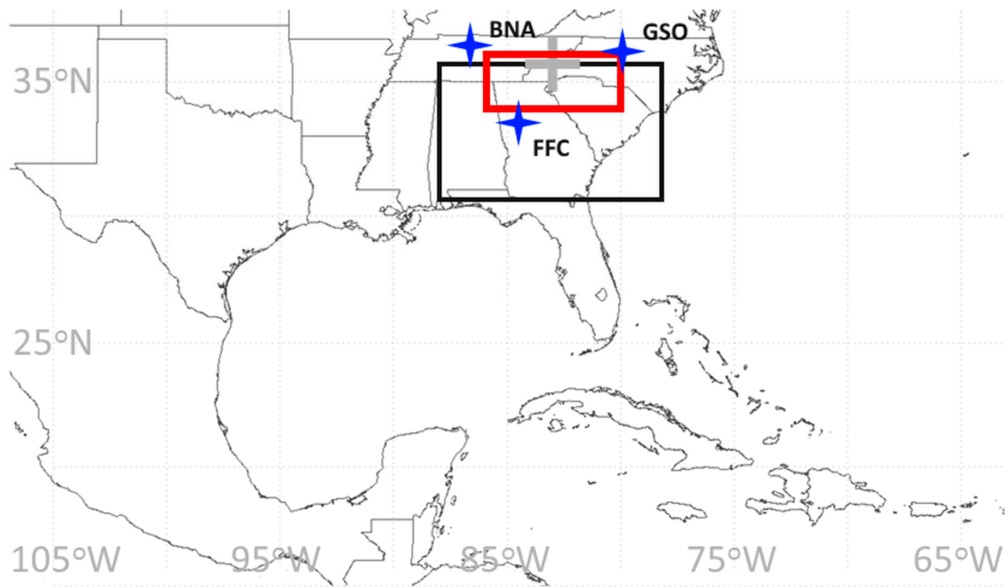

**Figure 2.** Study domain utilized for exploring Global Forecast System-analyzed fields of vertically-integrated horizontal water vapor transport xtending $5°$ south and $5°$ east and west from the PRB center point ($35.5°$ N, $83.0°$ W, elevation of 892 m above sea level, highlighted by a dark gray '+' symbol) is highlighted in black. An abbreviated study domain utilized for the sounding/20CR AR detection algorithm is highlighted in red. Locations of the operational Nashville, Tennessee (BNA), Greensboro, North Carolina (GSO), and Peachtree City, Georgia (FFC) upper-air stations are highlighted at the center of the blue four-point stars.

### 2.2. Examination of ARs in the 'Distant' Past (4 June 1936–30 June 2017)

Archived upper-air sounding observations at Peachtree City, Georgia (FFC), Greensboro, North Carolina (GSO), and Nashville, Tennessee (BNA), located in close proximity to the IVT study domain (Figure 2), were utilized to create an IVT climatology and compute 12-h IVT anomalies dating as far back as 1936 for the latter two stations. Unfortunately, the FFC observing station only became operational in September 1994 and limited the examination of ARs to a more recent 'distant' past.

Gridded data of the Twentieth Century Reanalysis (20CR) Project [33] over a significantly reduced IVT study domain (red boundary outline of Figure 2), consisting of eight grid points ($2° \times 2°$ latitude and longitude horizontal grid spacing of the 20CR) extending from 34 to $36°$ N and 80 to $86°$ W, was used to compute the 1000–700 hPa layer wind climatology (1 January 1994–31 December 2014) and 6-hourly wind anomalies over the recent 'distant' past (1 September 1994–31 December 2014) as a measure of low-level (jet) baroclinic processes near the Duke GSMRGN and CHLRGN.

We fine-tuned the new sounding/20CR-based algorithm such that an AR was detected if (1) the IVT anomaly at FFC or GSO fell within the top 5%, (2) the IVT anomaly at FFC or GSO fell within the top 10% and the 20CR low-level jet anomaly within the sounding/20CR IVT domain fell within

the top 16% (moderately strong low-level baroclinic processes), or (3) the average ranking of the IVT anomaly at FFC or GSO and the 20CR low-level jet anomaly within the study domain fell within the top 10% (strong low-level baroclinic processes). In the latter category, a strong 20CR low-level jet anomaly can offset a relatively 'weak' IVT anomaly (e.g., top 19%) to flag a weather event associated with an AR. Initiation or conclusion of an AR event corresponded to the period when consecutive 6-hourly top 25% 20CR low-level jet anomalies overlapping an IVT anomaly (meeting one of the three criteria above) increased from or dropped to a significant relative minimum, respectively. In some instances, primarily during the warm season, criterion (1) was met with a single sounding-based IVT anomaly, without significant (top 25%) 20CR low-level jet anomalies, in which case the AR duration was 12 h; at the 6-h synoptic period started at the time of the sounding observation plus the next 6-h period. The new sounding/20CR AR detection algorithm was focused on a much narrower corridor of influence by ARs than that of Miller et al. [5], but was also more lenient on weather events qualifying as AR-influenced as there is no shape or period-of-influence criterion. It must be acknowledged that all peer-reviewed AR detection algorithms contain a degree of subjectivity and that results based on a particular algorithm may change slightly if the methodology of a different peer-reviewed detection algorithm is followed. As nicely summarized in Mahoney et al. [22], a definition of what constitutes significant horizontal vapor transport is likely geographically and seasonally dependent, making a universal AR detection algorithm rather challenging. For this reason, our new detection algorithm was applied to a specific region and would require retuning IVT and low-level jet anomaly thresholds qualifying as significant in regions outside of the southeastern U.S.

Because landslides can be initiated during extended periods of above average rainfall [30], Elevated Rain Time Clusters (ERTCs) were calculated for the period of the 'distant' past (since 4 June 1936) using CHLRGN rainfall observations binned to the 6-h synoptic periods. Rain-free periods of $N \times 6$-h (where $N = 1$ to 16 synoptic periods) were counted for the total period of CHLRGN record (starting 4 June 1936). Slightly more than half of all rain-free periods were of at least five consecutive 6-h synoptic periods of duration. Analogous to the binning of rain gauge observations for single rainfall events described previously, rainfall amounts recorded at or after the start of a prolonged period of rain were counted as a single ERTC until the final amount of rain was recorded at least five 6-h synoptic periods (30-h) before the next amount was recorded in the CHLRGN. Rainfall accumulation at each gauge during each ERTC was summed and the total divided by the number of reporting gauges, giving the per gauge accumulation which was ranked to identify extreme (top 2.5%) ERTC events.

## 3. Results

### 3.1. Reconstructing ARs for the Study Domain

A comparison of ARs detected using the sounding-based IVT anomalies alone to the GFS-based methodology of the 'recent' past (after Miller et al. [5]) over the eight-year study period revealed ambiguities, particularly in summer and early autumn weather scenarios, requiring information independent of upper-air sounding observations. Close inspection of the combined sounding/20CR-based technique for detecting ARs compared to the GFS-based methodology of Miller et al. [5] over a five-and-a-half-year period (1 July 2009–31 December 2014, termination point of the current 20CR v2c data set) revealed the following patterns. Of 33 ARs detected by the new algorithm not flagged by the GFS-based technique, all but three ARs were qualified either as weak (minimum threshold falling below 500 kg m$^{-1}$ s$^{-1}$) or as not meeting the shape criterion as defined in Mahoney et al. [22] or as not impacting the Miller et al. [5] IVT study domain for at least 12 h. The three AR new algorithm 'false detections' (19 July 2010, 29–30 August 2010, 18 June 2013) were warm season events that, in two cases, were associated with an east-west oriented stationary front located near the southeastern U.S. Of 37 ARs detected by the GFS-based technique not flagged by the new algorithm, all but three 'missed' ARs tracked and/or intensified on the periphery of the Miller et al. [5] IVT study domain, outside the influence of the smaller sounding/20CR algorithm IVT domain (red border in

Figure 2). Two of the three 'missed' ARs tracking over the sounding/20CR IVT domain qualified as very weak while overhead the abbreviated domain (19 August 2012 and 5 April 2014), and one event (28 December 2014) tracked rapidly across the small domain, essentially 'missed' by the coarse 12-h time resolution of the operational soundings.

*3.2. Atmospheric Rivers and Rainfall Events Impacting Two River Basins in the Southern Appalachian Mountains during the 'Recent' Past (1 July 2009–30 June 2017)*

Over the eight-year period, 192 AR events, totaling 1294 6-h periods, were detected out of 11,463 possible 6-h periods, or 11.3% of the time during the eight years study period rain was falling owing its origin to ARs (Table 1). Seasonally, winter had the greatest and summer the fewest number of ARs. AR events in the summer and autumn were longer in duration than those in other seasons, primarily due to the tendency of cut-off lows to form and drift slowly eastward in the deep south of the U.S. during the warm months of summer and autumn. Extreme events (top 2.5%) occurred 43 and 24 times in the PRB and CRB, respectively (Table 2). The significantly greater number of extreme events of the Duke GSMRGN was due to the shorter record length compared to the CHLRGN, over which the 2.5% was calculated. Of the AR-, tropical cyclone- (TC-), or other-influenced weather scenarios contributing to extreme rain events, a majority of the extreme rain events in both river basins were influenced by an AR. ARs influenced a significant majority of the extreme events observed in the CRB over the eight-year period.

The average duration of AR events (Table 1) is relatively long compared to some other studies (e.g., Reference [21]). There are three issues contributing to the longer average duration of our study using the GFS-based AR detection algorithm. First, an AR was considered 'active' in our study if it covered any portion of the IVT study domain (Figure 2), which has a zonal width of ten degrees in longitude (centered on 83° W) and a meridional extent of five degrees in latitude (south of 35.5° N). Second, the imposed 12-h minimal threshold required for an event to be categorized as an AR in our study eliminated fast-moving events that may have contributed to lower average durations in other studies. Third, cool season extratropical storms generating ARs in the southeastern U.S. were at or rounding the upper-level trough as they transited through the IVT study domain. Hence, the storms often were pivoting, transitioning from eastward- to northeastward-propagating, while their ARs were covering the central and eastern half of the IVT study domain. Warm season extratropical storms generating ARs in the southeastern U.S. were slow-moving due to the equivalent barotropic characteristics of the environment and located far to the south of the primary baroclinic zone and jet stream.

**Table 1.** Seasonal count of atmospheric rivers impacting the vertically-integrated horizontal water vapor transport study domain using available 11,463 GFS 6-h analysis periods of the 8-y study (of 11,688 maximum 6-h periods; 98.1% of a complete archive).

| Meteorological Season | AR Events | 6-h Periods | Average Duration (h) |
|---|---|---|---|
| Winter (DJF) | 79 | 451 | 34.25 |
| Spring (MAM) | 49 | 258 | 31.59 |
| Summer (JJA) | 24 | 175 | 43.75 |
| Autumn (SON) | 40 | 247 | 37.05 |
| Total/average | 192 | 1294 | 35.34 |

**Table 2.** Number of extreme (top 2.5%) rain events of the eight-year study [1 July 2009–30 June 2017] {top two rows} and of the 20+ year study [1 September 1994–31 December 2014] {bottom row}.

| River Basin | AR-Influenced | TC-Influenced | Not AR- or TC-Influenced | Total |
|---|---|---|---|---|
| Pigeon | 23 (53.5%) | 3 (7.0%) | 17 (39.5%) | 43 |
| Coweeta | 20 (83.3%) | 2 (8.3%) | 2 (8.3%) | 24 |
| ---------- | ---------- | ---------- | ---------- | ---------- |
| Coweeta | 50 (58.8%) | 20 (23.5%) | 15 (17.7%) | 85 |

A comparison of extreme events occurring in the two river basins over the eight year period showed some disagreement. Extreme events in the PRB did not also qualify as extreme events in the CRB, due to record length differences and the calculation of "extreme" over those periods of record. Seven of the 24 extreme events in the eight year period qualified as extreme in the CRB, but did not qualify as extreme in the PRB. All events were associated with an AR. Composites of six of the seven extreme events unique to the CRB are plotted in Figures 3 and 4. One event (26 October 2010) involved a strong squall line that may have had a significant localized effect on the observed rainfall in both river basins and was omitted from further consideration.

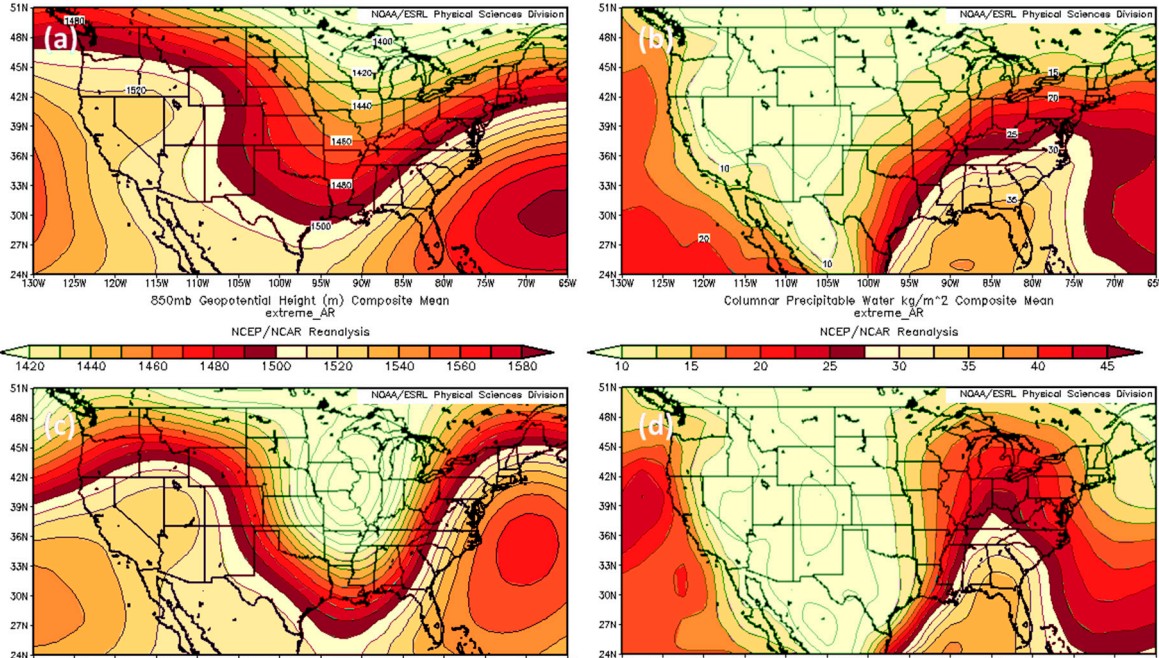

**Figure 3.** National Center for Environmental Prediction/National Center for Atmospheric Research (NCEP/NCAR) Reanalysis-based [45] composite mean of AR-influenced (**a**) 850 hPa geopotential height [m] and (**b**) precipitable water [kg m$^{-2}$] for 15 events in the PRB of the [5] Miller et al. (2018) study and AR-influenced (**c**) 850 hPa geopotential height [m] and (**d**) precipitable water [kg m$^{-2}$] for six events producing extreme rainfall events in the CRB and ordinary rainfall observed in the PRB. [Courtesy of NOAA's Earth System Research Laboratory].

Although differences in composite anomaly magnitudes (Figure 4c,d) may not be reliable owing to the relatively small sample ($N = 6$) used in creating the CRB composites, differences in orientation are significant. Composites of AR-influenced events in which ordinary precipitation was experienced in the PRB and extreme precipitation was experienced in the CRB (Figures 3c and 4c) shows a significantly higher amplitude, neutrally-tilted trough and ridge compared to the composites of AR-influenced events in which extreme precipitation was experienced in the PRB (Figures 3a and 4a). Differences in orientation of the large-scale wave imply different angles of incidence between the ARs, their associated low-level airflow, and local topography. Taken in their entirety, Figures 3 and 4 suggest AR scenarios having southwesterly flow favor extreme rainfall events in the PRB, while AR scenarios having southerly flow inhibit them. Hence, extreme rainfall in the PRB appears possible only under specific conditions due to the rain-shadowing effect caused by its horizontal distance north and west of the Blue Ridge Escarpment and to the localized topography bordering the basin, particularly to the south of the PRB.

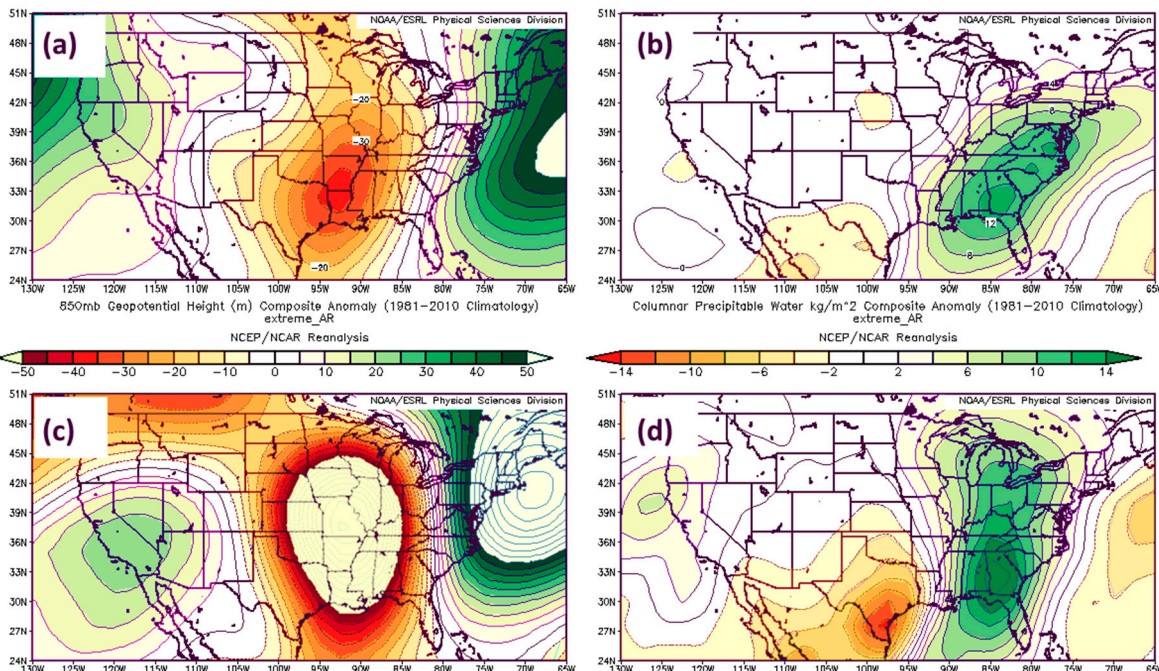

**Figure 4.** As in Figure 3, except NCEP/NCAR Reanalysis-based composite anomaly fields of AR-influenced (**a**) 850 hPa geopotential height [m] and (**b**) precipitable water [kg m$^{-2}$] for 15 events in the PRB of the [5] Miller et al. (2018) study and AR-influenced (**c**) 850 hPa geopotential height [m] and (**d**) precipitable water [kg m$^{-2}$] for six events producing extreme rainfall events in the CRB and ordinary rainfall observed in the PRB [Courtesy of NOAA's Earth System Research Laboratory].

*3.3. Rainfall Events (4 June 1936–30 June 2017) and Atmospheric Rivers (1 September 1994–31 December 2014) Impacting the Coweeta River Basin*

Extreme events of the CRB were most likely to occur in the autumn season and least likely in the summer season (Table 3). However, the summer extreme events were longer and had greater event totals of the network than other seasons. As noted in the shorter analysis, long-duration extreme events in the warm seasons of summer and early autumn were typically linked to a cut-off low located west of the southern Appalachians, displaced far from the baroclinic zone and jet stream in Canada.

**Table 3.** Seasonal extreme (top 2.5%) rain events observed by the Coweeta Hydrologic Laboratory gauge network coincident with 6-h analysis periods of the 82-year study [4 June 1936–31 December 2018].

| Meteorological Season | Rain Events | 6-h Rain Periods | Total per Gauge Accumulation (mm) | Average Duration (h) | Event Average per Gauge Accumulation (mm) |
|---|---|---|---|---|---|
| Winter (DJF) | 59 | 581 | 7697.71 | 59.08 | 130.47 |
| Spring (MAM) | 50 | 417 | 6349.67 | 50.04 | 126.99 |
| Summer (JJA) | 43 | 512 | 7004.51 | 71.44 | 162.90 |
| Autumn (SON) | 89 | 801 | 12,616.91 | 54.00 | 141.76 |
| Total/average | 241 | 2311 | 33,668.80 | 58.64 | 139.70 (5.5 in) |

AR totals of both study periods and AR detection methods, shown in Tables 1 and 4, averaged about 24 events per year. The extended period of the CRB examination showed consistency with the eight year study period in that ARs were found to occur most frequently in the winter season and least frequently in the summer season. The relatively short duration of the CRB summer season ARs was due to the equivalent barotropic nature of the environment of the cut-off lows in which they were embedded, with weak low-level winds and baroclinic processes, the metric used based on the 20CR 1000–700 hPa layer wind fields to fix AR event duration. Because of the seasonal mismatch in extreme rainfall and ARs displayed in Table 2, only 58.8% of extreme rainfall events were AR-influenced (bottom row) during the 20+ year study period. This percentage is a significant drop from the eight

year study period (top two rows, Table 2) and is attributed to the high occurrence of landfalling tropical cyclone (TC) activity in the region during the 1998–2007 decade and corresponding influence on extreme rainfall events (Figure 5). Aside from the drop in number of events corresponding to the strong El Niño episode of 1982–1983, extreme rainfall events of all influences (ARs, landfalling TCs, others) in the CRB have shown a steady increase over the past 82 years. No trend is obvious in the 11-year running mean of extreme rainfall events in the CRB when landfalling TC influences are removed (cf. orange and blue lines in Figure 5).

**Table 4.** Seasonal count of number of ARs impacting the smaller sounding/20CR IVT study domain using available 29,707 20CR 6-h analysis periods of the 20+ year study [1 September 1994–31 December 2014].

| Meteorological Season | AR Events | 6-h Periods | Average Duration (h) |
|---|---|---|---|
| Winter (DJF) | 151 | 919 | 36.52 |
| Spring (MAM) | 132 | 756 | 34.36 |
| Summer (JJA) | 55 | 185 | 20.18 |
| Autumn (SON) | 137 | 818 | 35.82 |
| Total/average | 475 | 2678 | 33.83 |

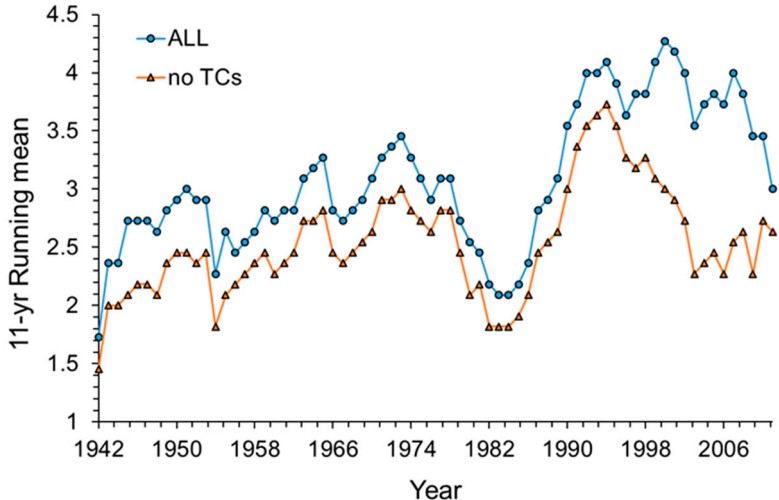

**Figure 5.** 11-year running mean of extreme rainfall events observed by the Coweeta Hydrologic Laboratory gauge network covering the period 1937–2016 (graph covers 1942–2011). Blue (orange) trace represents the total number of extreme rainfall events (not associated with landfalling tropical cyclones).

*3.4. Connections between Rainfall, Atmospheric Rivers, and Landslides in the Southern Appalachian Mountains*

The most active years for landslide days in western North Carolina were 2003, 2005, 2009, and 2013, in ascending order, while few landslide days occurred over an extended period from 1997 through 2002 (Figure 6). Correlations among AR events associated with extreme ERTCs, extreme ERTCs, or extreme rainfall events (ExtRs) and landslide days (LDSL) were low, and insignificant, with Pearson correlation coefficients ranging from r = 0.09 (AR vs. extreme ERTC) to r = 0.27 (AR vs. LDSL, Table 5). In general, extreme ERTC events were linked with at least one extreme rainfall event in the CRB during the period of the ERTC (r = 0.62, Table 5) throughout the 21-year study period. It is clear when comparing the AR series with those of the other parameters over the 20+ year period (Figure 6) that 2007–2008 represented a transition from ARs tracking with an opposite trend to LDSL, extreme ERTC, and ExtR events in the earlier period, to a change of AR trend being in phase with the trends of other events during the more recent post-2007–2008 period. Western North Carolina experienced a significant and extended drought during the 2007–2008 transition period [46], which also corresponded to the period of a strong La Niña [47] and strongly positive phase of the Pacific North American weather pattern toward the end of 2007 [48]. Global and hemispheric weather patterns

undoubtedly played some role in controlling AR activity influencing the southern Appalachians during the 20+ year study period, but their exact role awaits future study.

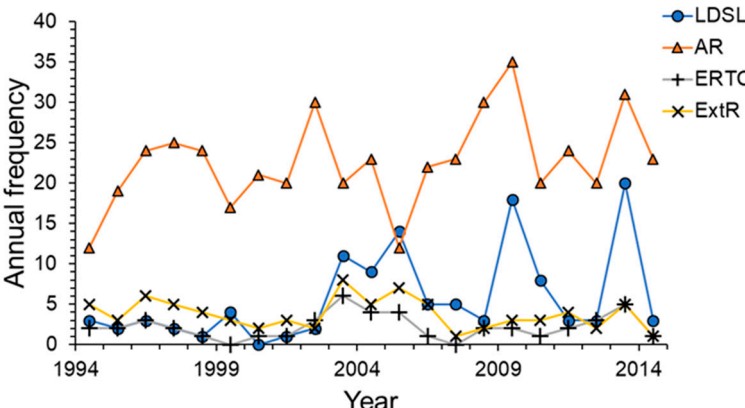

**Figure 6.** Annual number of AR (orange '**Δ**'), landslide day (LDSL, blue '**O**'), extreme elevated rain time cluster (ERTC, gray '**+**'), and extreme rainfall (ExtR, gold '×') events from 1994 to 2014.

**Table 5.** Pearson correlation coefficient (top row) and *p*-value (bottom row) of LDSL, AR, extreme ERTC, and ExtR events from 1994–2014. Those linked events having a relatively high correlation and low *p*-value are highlighted in boldface font.

|  | LDSL | AR | Extreme ERTC | ExtR |
|---|---|---|---|---|
| **LDSL** | X | 0.273 <br> 0.231 | **0.561** <br> **0.008** | 0.406 <br> 0.068 |
| **AR** | 0.273 <br> 0.231 | X | 0.093 <br> 0.689 | −0.283 <br> 0.214 |
| **Extreme ERTC** | **0.561** <br> **0.008** | 0.093 <br> 0.689 | X | **0.672** <br> **0.001** |
| **ExtR** | 0.406 <br> 0.068 | −0.283 <br> 0.214 | **0.672** <br> **0.001** | X |

　　　Landslide days were clustered in time (Figure 7); calendar year 2003 was active early (six in March–May) and late (five in October–December) with the early period events linked via soil pre-conditioning and/or initiation to simultaneous ARs, extreme ERTCs, and ExtRs and the late period seemingly driven primarily by ExtRs. Landslide days in 2005 were linked primarily to landfalling TCs, extreme ERTCs, and ExtRs, with a noticeable dearth of ARs (Figure 6), a period of transition from a weak El Niño to a weak La Niña [47]. The number of extreme ERTCs in 2009 (and early 2010) was rather low, so most landslide days of that year (and early 2010) were conditioned or triggered either by individual strong ARs or ExtRs (Figure 7). Calendar year 2013 was notable for significant rainfall in western North Carolina [5]. Numerous landslide days of that year were linked to simultaneous ARs, extreme ERTCs, and ExtRs, with several of the heaviest rainfall events (January, May, and July) consisting of multiple ARs within a single extreme ERTC. The most extreme non-tropical-influenced ERTC of the 46 events over the 21 (or 20+) year study period occurred in July 2013 and was associated with seven landslide days (Table 6). It must be noted over the study period that landfalling TCs were responsible for the greatest number of landslides in a day, even though their period of influence was brief (low number of landslide days).

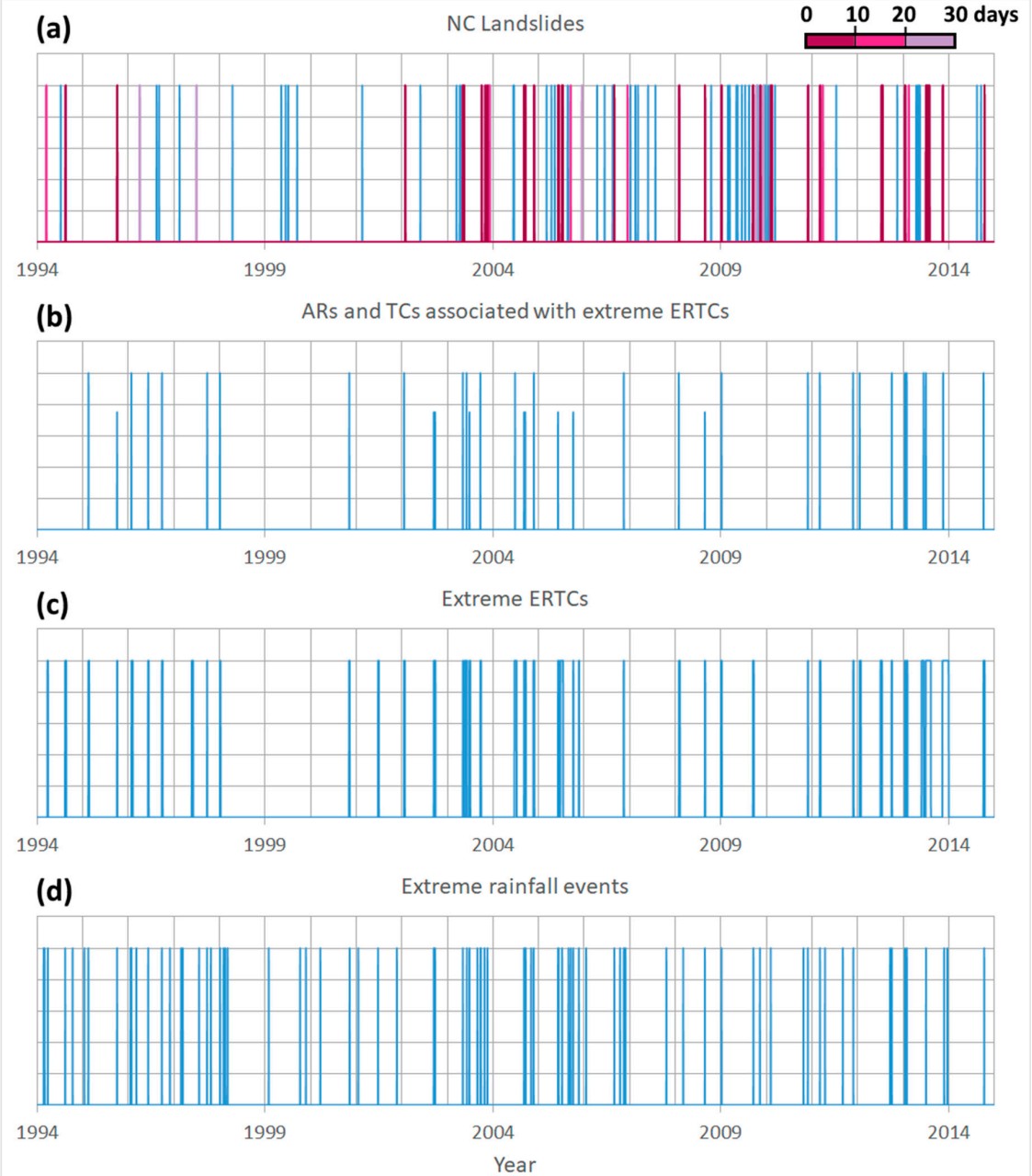

**Figure 7.** Time series marking events of (**a**) landslide days, (**b**) ARs and landfalling TCs [if any] associated with extreme ERTCs, (**c**) extreme ERTCs, and (**d**) occurrences of extreme rainfall over the 21 year study period from 1 January 1994–31 December 2014. Events of panels (**b**)–(**d**) occurring within 10, 10–20, and 20–30 days of a landslide day (scale in upper right corner) have landslide day marks plotted in deep purple, medium purple, and light purple, respectively, in panel (**a**). The amplitude of marks in panels (**a**), (**c**), and (**d**) is meaningless as marks are meant merely to represent the starting day when a particular event was "on." Short marks in panel (**b**) represent landfalling TCs, while full-sized marks represent ARs.

**Table 6.** Top 15 of the 46 extreme (top 2.5%) Elevated Rain Time Cluster (ERTC) events observed by the CHLRGN over the 20+ year study ranked according to total per gauge accumulation. Extreme rainfall (ExtR)-, AR-, or TC-influenced events, are recorded in Column 7, a number in brackets represents the number of ARs influencing a given ERTC (one is the default). The entry in Column 7 represents the dominant large-scale weather influence, if any exists, and a '+' indicates an ExtR event also coincided with the dominant influence. A day having a reported landslide or landslides in western North Carolina occurring within 30 days of the ExtR, AR, or TC is recorded in Column 8, a number in brackets represents the number of landslide days linked to a given ERTC (one is the default). The maximum number of reporting gauges during all 6-h synoptic periods of an event are recorded in Column 9.

| Rank | Starting | | | | Duration (h) | ExtR, AR, and/or TC [#] | Landslide(s) [# days] | Max. No. of Reporting Gauges |
| | Year | Month | Day | Hour (UTC) | | | | |
|---|---|---|---|---|---|---|---|---|
| 1 | 2004 | 9 | 16 | 0600 | 114.0 | TC+ | Yes [2] | 7 |
| 2 | 2013 | 7 | 1 | 0600 | 1002.0 | AR [3]+ | Yes [7] | 8 |
| 3 | 2013 | 11 | 12 | 1200 | 1044.0 | AR [7]+ | Yes | 8 |
| 4 | 2008 | 8 | 25 | 0600 | 66.0 | TC+ | Yes | 9 |
| 5 | 2005 | 6 | 26 | 0000 | 510.0 | ExtR | Yes [4] | 7 |
| 6 | 2003 | 6 | 27 | 1200 | 132.0 | TC+ | No | 8 |
| 7 | 2005 | 11 | 20 | 0000 | 72.0 | ExtR | No | 4 |
| 8 | 2005 | 6 | 6 | 0000 | 180.0 | TC+ | Yes [3] | 6 |
| 9 | 2009 | 9 | 15 | 1200 | 162.0 | ExtR | Yes [2] | 8 |
| 10 | 1996 | 9 | 26 | 0000 | 186.0 | AR+ | No | 9 |
| 11 | 1995 | 10 | 3 | 0600 | 60.0 | TC+ | Yes [2] | 9 |
| 12 | 2013 | 1 | 14 | 0000 | 96.0 | AR [2]+ | Yes [4] | 8 |
| 13 | 2012 | 7 | 2 | 0000 | 330.0 | N/A | Yes | 8 |
| 14 | 2014 | 10 | 6 | 1800 | 228.0 | AR [3]+ | Yes | 8 |
| 15 | 1996 | 1 | 26 | 1200 | 228.0 | AR+ | No | 9 |

Seasonally, extreme ERTCs and rainfall events (both event types are often linked) as observed by the CHLRGN were most likely to occur in autumn, while ARs peaked in the winter, and the number of landslide days was greatest in the summer, although the low range of the latter event type indicates a nearly even distribution across all seasons (Table 7).

**Table 7.** Seasonal count of landslide days, atmospheric rivers, extreme (top 2.5%) Elevated Rain Time Clusters (ERTC), and extreme (top 2.5%) rainfall events over the 21-year study (or 20+ year study for ARs) [1 January 1994–31 December 2014]. The latter two event types were counted using rainfall observations of the CHLRGN.

| Meteorological Season | Landslide Days | Atmospheric Rivers * | Extreme ERTCs | Extreme Rainfall Events |
|---|---|---|---|---|
| Winter (DJF) | 28 | 151 | 9 | 17 |
| Spring (MAM) | 30 | 132 | 6 | 10 |
| Summer (JJA) | 36 | 55 | 11 | 14 |
| Autumn (SON) | 26 | 137 | 20 | 38 |
| Total | 120 | 475 | 46 | 79 |

* The period of record for ARs starts in September 1994 corresponding to the time that the Peachtree City, Georgia upper-air station became fully operational.

Combinations of event types contributing to the pre-conditioning or initiation of landslides are summed in Figure 8 over the 20+ year study period during which 117 landslide days occurred. Landslide days were correlated with extreme ERTCs (Table 5, r = 0.56, $p < 0.01$), but when ARs, ExtRs, and extreme ERTCs coincided, we observed the greatest frequency of landslide days (21.4%). Landslide initiation and/or conditioning by ARs alone (18 LDSL, 15.4%) and by "Other" (19 LDSL, 16.2%) were the second highest contributors to the total number of landslide days of Figure 8, but not significantly correlated as noted above. Several above-normal ERTC events and above-normal rain events contributed to a total of 16 landslide days. While significantly correlated, an extreme ERTC was not necessary for a landslide to occur.

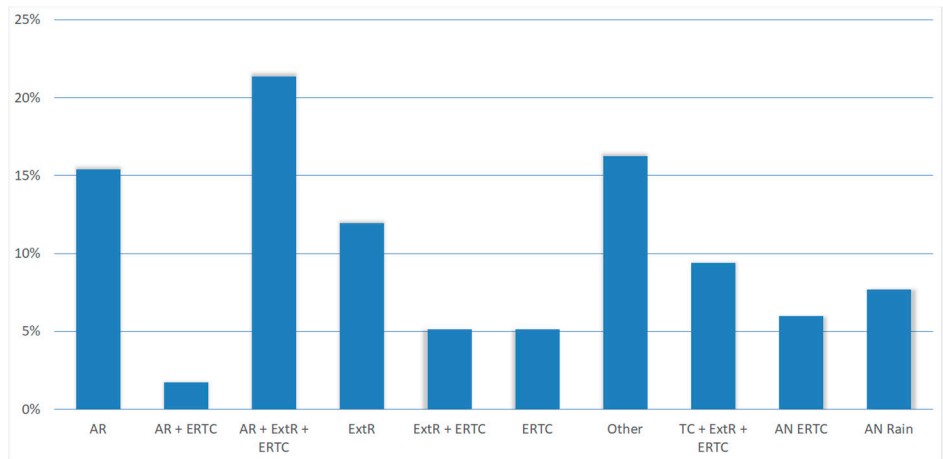

**Figure 8.** Percentage of 117 landslide days over the 20+ year study period associated with individual or combinations of atmospheric river (AR), landfalling tropical cyclone (TC), extreme rainfall (ExtR), and extreme elevated rainfall time cluster (ERTC) events. 'AN' of the final two bars represents 'above normal' (at least top 33%, but not in the top 2.5%) ERTC and rainfall events.

Individual contributions of AR, ExtR and extreme ERTC event types to landslide days (Figure 9a–c) show nearly identical frequencies, with 38% of landslide days (45 events) traced to AR- and ExtR-influenced events and 33% of landslide days (39 events) traced to extreme ERTC-influenced events. Coincident AR, ExtR, and extreme ERTC events accounted for 25 landslide days (Figure 8), coincident AR and extreme ERTC events accounted for two landslide days, and coincident ExtR and extreme ERTC events accounted for six landslide days. Close inspection of the 25 landslide days found they corresponded to eight cases of coincident AR, ExtR, and extreme ERTC event types during the study period. Nine cases of the three coincident event types resulted in zero landslide days. Although rare, extreme ERTC events coupled with multiple occurrences of ARs or ExtRs during the period of the extreme ERTC event resulted in landslide days in four of the six cases. Recall that a single AR, ExtR, and/or extreme ERTC event was sometimes linked to multiple landslide days. As might be expected, frequencies of ARs and ExtR influencing landslide days are much lower than their influence of flooding events in the PRB found by Miller et al. [5] (cf. Figure 3d,e), ~64% and 50%, respectively, as might be expected since many factors, in addition to rainfall, are important for initiating landslides. Of the 475 AR, 79 ExtR, and 46 extreme ERTC events detected during the 20+ year study, 32 (6.7%), 30 (38.0%), and 29 (63.0%), respectively, played a role in at least one landslide day.

Aside from pre-conditioning western North Carolina for landslides via moistening of the soil, a second method for AR, ExtR, and extreme ERTC events to pre-condition the environment is through modification of the large-scale weather pattern. One example of this can be seen in the extreme ERTC event of July 2013 (ranked #2 in Table 7, [49]) that showed a high amplitude wave at the mid- and upper-troposphere (Figure 10a,b), which is an unusual weather pattern for the middle of summer. The center of the Bermuda High at the 850 hPa level (Figure 10c) was located well offshore of the U.S. east coast and a corridor of air having high precipitable water values bordered the western edge of the Bermuda High (Figure 10d). An anomalously strong ridge over Maine and Nova Scotia at the upper- and mid-troposphere (Figure 11a,b) was also evident. The trough geopotential height anomaly at the upper- and mid-troposphere located over the Mississippi River Valley was half as strong as the ridging to the northeast. The low-troposphere geopotential height anomalies at the 850 hPa level (Figure 11c) were indicative of an unusually strong southeasterly, low-level airflow. The anomaly in precipitable water (Figure 11d) had an unusually high available moisture bullseye to the north, just offshore of Maine and Nova Scotia, with a narrow corridor of higher-than-normal available moisture connecting the bullseye southward to the Gulf of Mexico.

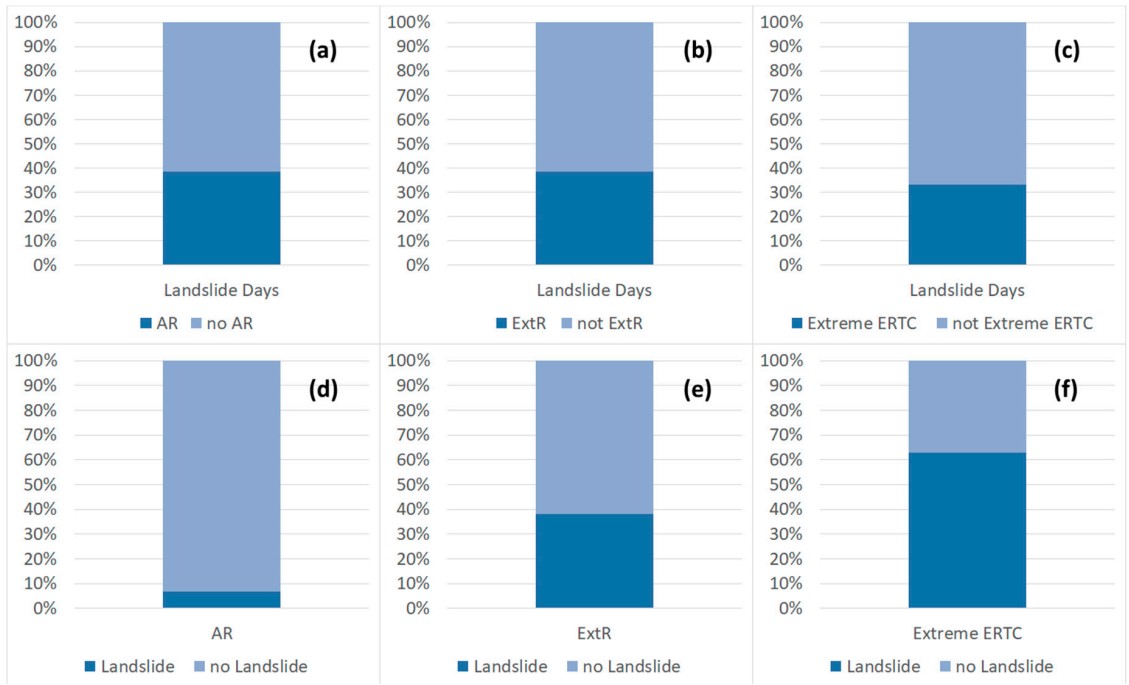

**Figure 9.** Frequency of (**a**) AR events, (**b**) extreme rainfall events, and (**c**) extreme ERTC events influencing landslide days of western North Carolina. Proportion of all (**d**) AR events [of 475], (**e**) extreme rainfall events [of 79], and (**f**) extreme ERTC events [of 46] influencing a landslide day over the 20+ year study period (1 September 1994–31 December 2014).

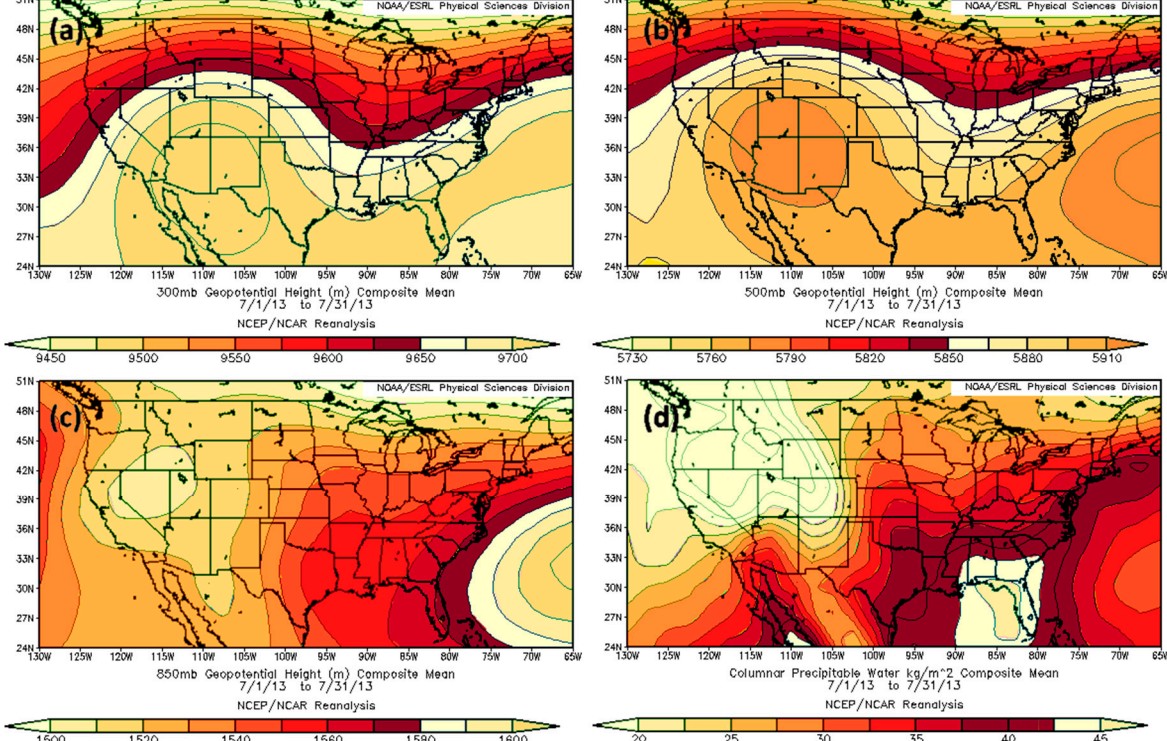

**Figure 10.** NCEP/NCAR Reanalysis-based [45] composite mean of (**a**) 300 hPa (**b**) 500 hPa, and (**c**) 850 hPa geopotential height [m] and (**d**) precipitable water [kg m$^{-2}$] for July 2013 [Courtesy of NOAA's Earth System Research Laboratory].

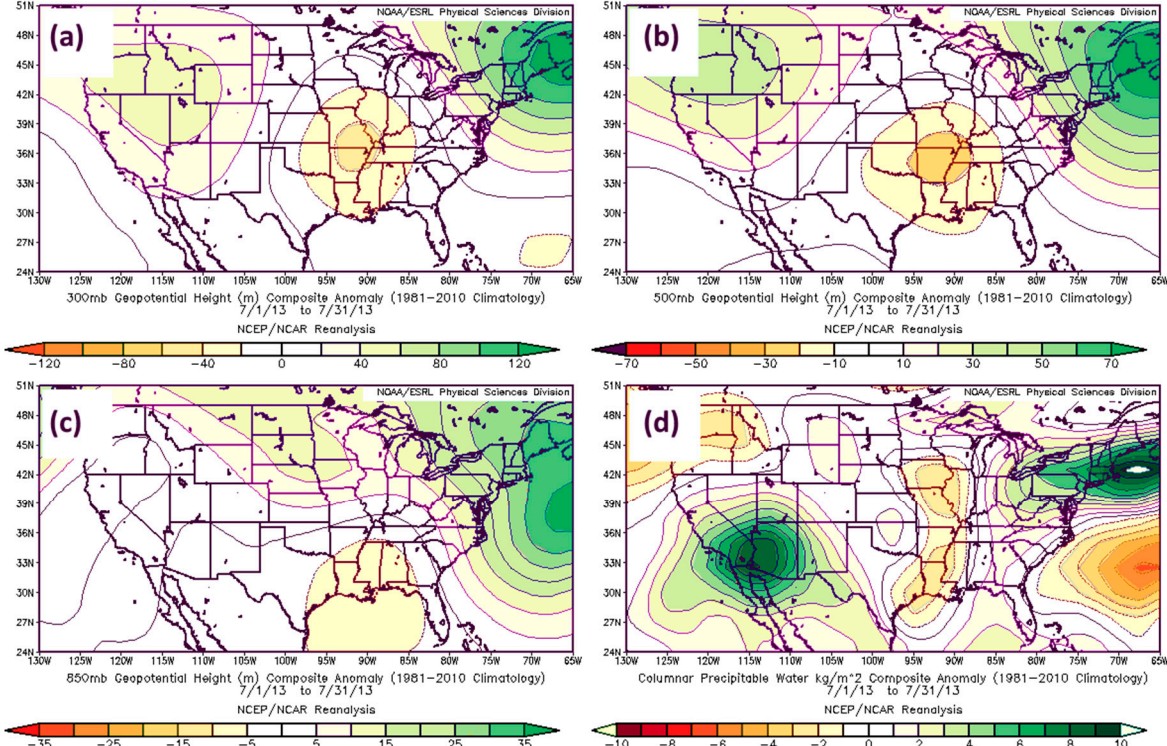

**Figure 11.** As in Figure 10, except NCEP/NCAR Reanalysis-based composite anomaly fields of (**a**) 300 hPa (**b**) 500 hPa, and (**c**) 850 hPa geopotential height [m] and (**d**) precipitable water [kg m$^{-2}$] for July 2013 [Courtesy of NOAA's Earth System Research Laboratory].

We suggest that the anomalously strong ridging to the northeast and the connection of this region to the Gulf of Mexico, as suggested by the precipitable water composite mean and anomaly, was made possible by a series of ARs that were observed at the end of June (Figure 12a) and at the beginning and middle of July 2013 (Figure 12b). The IVT plumes (Figure 12) represent two of four strong ARs detected between 29 June and 16 July 2013 that followed nearly identical pathways as they moved north and east, along the western periphery of the Bermuda High. We propose that the anomalously strong ridge to the northeast was a consequence of ridge-building provided via the diabatic process of latent heat release as the sub-tropical moisture of the ARs was converted to cloud water and precipitation. As the ridge continued to build, the large-scale weather pattern became increasingly stagnant, allowing for the repeated propagation of successive ARs over the same pathway and increased ridge-building through continued latent heating. Hence, the stagnation of the large-scale weather pattern due to the ridge-building to the north via latent heat release, provided a mechanism for 'training' rainfall events over the southern Appalachian Mountains that pre-conditioned the soil and/or triggered landslides. This particular example of 'training' large-scale rainfall events was responsible for seven landslide days in July 2013 (Table 6). Following the 'Maya Express' nomenclature of Dirmeyer and Kinter [50], which implies a dynamic system, it is proposed that the large-scale weather mechanism describing the July 2013 event be designated as a 'Maya Corridor.' As a technology corridor describes a fixed zone along an interstate, a 'Maya Corridor' implies a fixed zone of significant meridional extent moving moisture from the subtropics into the mid-latitudes of the eastern U.S., making conditions favorable for flooding and landslides. Based on this example, AR, ExtR, and extreme ERTC events, under favorable large-scale patterns, are capable of providing the ridge-building necessary to halt the propagation of the large-scale wave and establish a 'Maya Corridor.' The third ranking extreme ERTC event of mid-November 2013 (Table 6) provides another example of a variation on the 'Maya Corridor' theme from the 20+ year study period involving seven ARs (not shown), but resulted in only a single landslide day, likely due to the relatively low background moisture typical of winter.

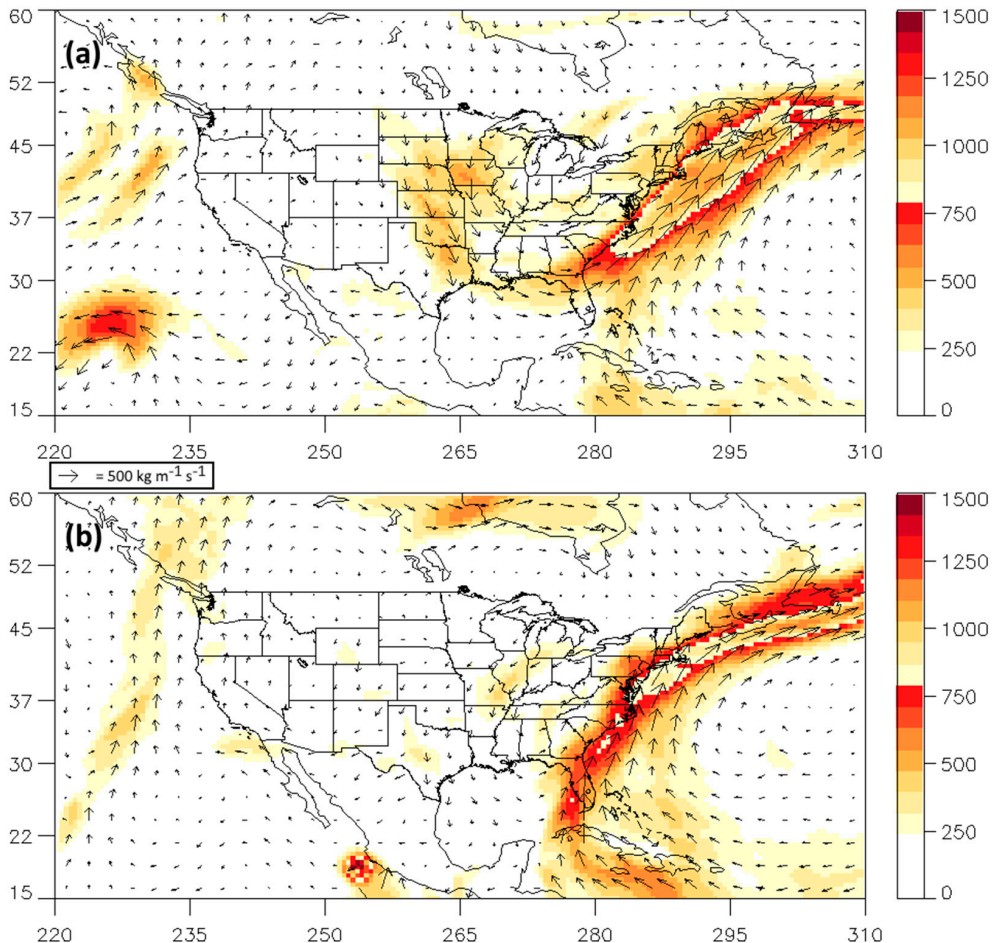

**Figure 12.** GFS-analyzed fields of IVT (kg m$^{-1}$ s$^{-1}$) valid at (**a**) 1800 UTC 29 June 2013 and (**b**) 1800 UTC 1 July 2013. Reference IVT vector is plotted just above panel (**b**).

## 4. Discussion

We detected and characterized ARs in the southern Appalachian Mountains in two river basins, located 60 km apart, and examined the influence of ARs on extreme rainfall (ExtR), periods of elevated precipitation (ERTC) and landslide (LDSL) events over two time periods, the 'recent' (for both basins) and 'distant' (CRB only) past. In the process of characterizing ARs, we developed a new method of AR detection that allowed reconstruction of an AR time series that extended back to 1994, allowing the recent record from 2009-to-present to be extended significantly. Our analysis showed that ARs account for the majority (53–59%) of extreme rainfall events compared to tropical cyclones and occur most often in winter and least often (but are longer-lasting) in summer. Impacts of ARs in terms of observed rainfall intensity differed between the two river basins during seven events over the 'recent' past, with the CRB experiencing extreme rainfall while the PRB did not. Lastly, we found that landslides were most correlated with elevated rain time clusters (ERTC), followed by extreme rainfall, followed by ARs.

A new technique for detecting ARs was tested so that extreme rainfall events of the long-record CHLRGN could provide context for several historic flooding and landslide events recorded in western North Carolina dating as far back as the mid-1900s. The sounding-based IVT anomalies, coupled with the 20CR low-layer jet anomalies over the smaller IVT domain, allowed the linkage of detected AR events to the earliest rainfall records of the CHLRGN overlapping with the sounding and 20CR archives. In tuning the sounding/20CR-based AR detection algorithm it became clear that BNA IVT anomalies too often gave a false detection as the AR would often weaken and, sometimes, disappear after having crossed the mountains. As noted in some investigations of warm season convection

(e.g., Reference [51]), lower tropospheric circulations often reorganize and re-intensify once the parent system has moved eastward from the mountains into the Piedmont (central) region of North Carolina. This notion is consistent with Lavers and Villarini [20] who found that the fraction of precipitation related to ARs over the continental U.S. to be a relative minimum over the Appalachians compared to the central and eastern U.S. Our additional criteria imposed on the data addressed these anomalies.

Our results utilized two long-term records (82-year and 20+ year) based on observations of the CHLRGN, a well-documented database of rainfall observations in the southern Appalachian Mountains (e.g., find References [52–54] as recent studies particularly relevant to this study incorporating CHLRGN observations), and showed remarkable consistency in seasonal distributions of extreme rainfall events (Table 3) and ARs (Table 4), compared to the shorter time series (Tables 1 and 2; cf. Tables 6 and 8 of Reference [5]). Both AR detection algorithms showed, on average, an annual count of 24 ARs, with the count of ARs reaching a maximum in the winter season and a minimum in the summer season. The number of ARs detected in the spring and autumn seasons was similar and over twice the number of those flagged in the summer. Trends in the 11-year running mean of extreme rainfall events in the CRB found a steady increase in the number of events associated with ALL influences (ARs, landfalling TCs, others [e.g., convection]) over the period dating back to 1937. The focus of this study was the impact of ARs on extreme precipitation and its potential consequences; landslides. Of interest in a future study is to document the average percentage annual rainfall owing its origin to ARs over the long-term CHLRGN record to investigate potential climatological trends. Recent studies (e.g., Reference [55]) have found links between decadal-scale baseflow variability of watersheds and oscillations (e.g., Atlantic Multidecadal Oscillation). The long-term CHLRGN record and extended AR data base developed in this study will allow future studies examining decadal variability of drought in watersheds of the southern Appalachian Mountains.

Over the recent eight-year period, seven AR-influenced events contributed to extreme rainfall in the CRB, according to the long-record CHLRGN archive, and to non-extreme rainfall in the PRB, according to the shorter record Duke GSMRGN archive. A composite of six of these AR-influenced events (the seventh was 'contaminated' by a squall line) suggested that ARs whose low-level flow was directed primarily from the south resulted in a muted precipitation response in the PRB. It was hypothesized that the distance north and west from the Blue Ridge Escarpment, in addition, to the relatively high elevation mountains lining the PRB at its southern border provided a rain-shadowing effect not evident in rainfall observations of the CRB for southerly flow. The operational implication of this finding is that a proposed AR severity scale, analogous to the enhanced Fujita scale for tornadoes, must account for local variations in topography and prominent wind direction of each AR when assigning a severity index for forecast zones located in mountainous regions. A modest eight-year study period revealed at least six examples when a severe AR index warning (e.g., 'flooding and flash flooding likely') might have verified in the CRB, but not verified in the PRB.

An examination of possible linkages between ARs, extreme rainfall events observed in the CRB (ExtR), elevated rain time cluster (ERTC) events in the CRB and landslides over a 20+ year study period showed that none of the event types were singlehandedly able to serve as a predictor for the occurrence of landslide days. However, an integrated approach showed that coincident occurrences of the event types gave some indicator of when landslide days were more likely. Of the 117 landslide days observed by the North Carolina Geological Survey during the 20+ year study period, the greatest number (25) were pre-conditioned or initiated when AR, ExtR, and extreme ERTC events occurred simultaneously (Figure 7), although this combination resulted in landslides only about half of the time. AR occurrence alone accounted for a significant number of landslide days (18), but the overall number of ARs during the study period (475) was quite large so using it alone as a landslide predictor was unreliable (Figure 9d). Of the 46 extreme ERTC events occurring during the 20+ year study period, 29 were linked to landslide days (Figure 9f). Individual extreme ERTC events, when linked with multiple ExtR and AR events or with multiple AR events (events ranks #2, #3, #12, and #14 of Table 6) during its occurrence showed potential as a landslide day forecast predictor worth pursuing in future

studies. Results of Tao and Barros [56] suggested seasonal differences in the time-scale response of landslides to pre-conditioning and/or initiating rainfall events, with the response being slower in the cool season. Therefore, the predictability of AR-, ExtR-, and/or extreme ERTC-related landslides may demonstrate significant skill during the cool season and is worthy of closer examination in future studies. The limit of the investigated AR, ExtR, and extreme ERTC parameters in predicting landslide days was shown in events when convective systems were responsible for landslide pre-conditioning and/or initiation (e.g., 14–15 July 2011). The small spatial and temporal scale of these events limited the ability of the sounding/20CR AR detection algorithm and the CHLRGN to "see" them. Additionally, it was found that several above normal (within the top 33%, but not in the top 2.5%) rainfall and ERTC events observed by the CHLRGN were likely conditioners and/or initiators of landslides (Figure 8). Hence, it is worth investigating in a future study if a relaxation of the threshold of rainfall and ERTC events from extreme to 'slightly less-than-extreme' might serve as better predictors of landslides without raising the number of false alarms.

Close inspection of the highest ranking extreme ERTC event having extratropical origins (event rank #2 in Table 6) showed an extraordinarily extended period of precipitation occurring in the middle of the summer season (July 2013), an event linked to seven landslide days. Multiple ARs occurred at the start of the extreme ERTC period and followed a fixed pathway along a corridor ('Maya Corridor') located west of the center of the Bermuda High. It is hypothesized that the 'training' synoptic-scale systems transiting along the U.S. east coast acted to amplify the upper-level ridge near Maine and Nova Scotia, through the diabatic process of latent heat release as vapor was converted to cloud water and precipitation. Once the 'Maya Corridor' was constructed by multiple ARs early in the period of the extreme ERTC, successive extratropical non-AR disturbances transited over the same pathway, depositing their rainfall over the same region of western North Carolina, thereby pre-conditioning and/or initiating subsequent landslides. Of interest in a future study will be to conduct a series of numerical model simulations over the period in late June and July 2013 using full (moist) and dry model physics to investigate differences in amplification of the upper-level downstream ridge associated with the Bermuda High.

Another future study will focus on extending the study period to the earliest records of the CHLRGN ('distant past'), investigating some of the linkages found in the recent 'distant' past (20+ year study period) between AR, ExtR, and extreme ERTC events and landslide days. Many noteworthy landslides occurred in western North Carolina from the 1940s through the 1980s and a linkage of these events to the occurrence of the three event types with a lead time of up to one month will serve as a test of their predictive capability. Investigation of a period before September 1994 will necessitate a recalculation of the sounding-based IVT climatology and 12-hourly IVT anomalies using observations from the now decommissioned Athens, Georgia upper-air observing station.

## 5. Conclusions

In conclusion, a comparison of two river basins located in the southern Appalachian Mountains showed differences in the extremity of rainfall observed in each when influenced by an atmospheric river, under specific conditions. In particular, the severity of observed rainfall in the Pigeon River Basin was suppressed when the low-level airflow of the AR was primarily from the south. It was surmised that the difference was due to a rain-shadowing effect resulting from its location being farther north and west of the Blue Ridge Escarpment than the Coweeta River Basin and the high elevation mountains lining its southern boundary.

An examination of extreme rainfall events in both river basins over a 'recent' (eight-year) and 'distant' (82-year) study period showed consistency in their seasonality. These events most likely occurred in the autumn season and were nearly as likely in the winter. However, extreme rainfall events in the summer had the longest average duration. Miller et al. [5] showed the warm season extreme events generally linked with a weather pattern consisting of a cut-off low in the deep south, a slow-moving feature displaced far from the primary jet stream axis located in southern Canada.

Analysis of the July 2013 period of extreme elevated precipitation in this study showed atmospheric rivers occurring in rapid succession early in the period, potentially contributing to building the downstream ridge, thereby stalling the large-scale weather pattern.

Although no single weather event type (atmospheric river (AR), extreme rainfall (ExtR), or extreme elevated rain time cluster (ERTC)) was found to be a perfect predictor of landslide days (LDSL), the latter showed a significant correlation with landslide days. Of the 29 extreme ERTC events occurring during the 20+ year study period, 63% were associated with at least one landslide day. Chances of a landslide day occurring during or within 30 days of an extreme ERTC increased to 66% if multiple AR and/or ExtR events were observed during the ERTC.

In summary, atmospheric rivers either pre-condition, through soil moistening, or initiate landslides, but also may pre-condition the large-scale weather pattern to a 'rainy' regime in the southern Appalachian Mountains by establishing a 'Maya Corridor' through diabatic heating caused by latent heat release of condensation and/or deposition as clouds and precipitation form in the downstream ridge. The stagnation of the large-scale weather pattern allows successive precipitation events to propagate over the same region, continuously moistening the soil and/or providing water that initiates landslide(s).

**Author Contributions:** Conceptualization, D.K.M.; Methodology, D.K.M., A.P.B., R.M.W. and C.F.M.; Formal Analysis, D.K.M.; Data Curation, R.M.W., A.P.B. and C.F.M.; Writing–Original Draft Preparation, D.K.M.; Writing–Review & Editing, R.M.W., A.P.B. and C.F.M.

**Funding:** Funding for this study was provided by a COMET GOES-R Partners Project grant number Z16-20569, NOAA-NESDIS (NC-CICS grant 2014-2918-08), F.M. Ralph at Scripps Institution of Oceanography, to D. Miller. NASA grants NNX07AK40G, NNX10AH66G, and NNX13AH39G, the Pratt School of Engineering at Duke University, to A. Barros at Duke University. Precipitation data at Coweeta Hydrologic Lab were funded by NSF grants DEB0218001, DEB0823293, DEB1440485, and DEB1637522, to the Coweeta LTER program at the University of Georgia and by USDA Forest Service, Southern Research Station, Coweeta Hydrologic Laboratory project funds.

**Acknowledgments:** The authors are grateful for the support and assistance of Paul Super of the National Park Service, land owners who have permitted the installation of a rain gauge on their property, University of North Carolina at Asheville and Duke University students, and Kyle, Hugh, Don, and Roger, support personnel of the Waynesville Watershed. We also gratefully acknowledge the helpful comments of several anonymous reviewers who greatly improved the quality of the manuscript. Color scales used in Figures 3, 4 and 10–12 were designed by Cynthia Brewer and Mark Harrower (http://colorbrewer2.org).

**Conflicts of Interest:** Any opinions, findings, conclusions, or recommendations expressed in the material are those of the authors and do not necessarily reflect the views of the USDA Forest Service. The authors declare that experiments complied with the current laws of the United States of America and there are no conflicts of interest.

## Appendix A

**Table A1.** Location and elevation of the 32 tipping-bucket rain gauges comprising the Duke GSMRGN and of the nine recording rain gauges (RRGs) comprising the CHLRGN.

| Duke GSMRGN Gauge Attributes | | | | | | | | CHLRGN Gauge Attributes | | | |
|---|---|---|---|---|---|---|---|---|---|---|---|
| Gauge | Lat. | Lon. | Elev. (m) | Gauge | Lat. | Lon. | Elev. (m) | Gauge | Lat. | Lon. | Elev. (m) |
| RG001 | 35°23.8′ | 82°54.7′ | 1156 | RG109 | 35°29.7′ | 83°02.4′ | 1500 | RRG06 | 35°3.62′ | 83°25.8′ | 687 |
| RG002 | 35°25.5′ | 82°58.2′ | 1731 | RG110 | 35°32.8′ | 83°08.8′ | 1563 | RRG05 | 35°3.63′ | 83°27.9′ | 1144 |
| RG003 | 35°23.0′ | 82°54.9′ | 1609 | RG111 | 35°43.7′ | 82°56.8′ | 1394 | RRG20 | 35°3.89′ | 83°26.5′ | 740 |
| RG004 | 35°22.0′ | 82°59.4′ | 1922 | RG112 | 35°45.0′ | 82°57.8′ | 1184 | RRG31 | 35°1.96′ | 83°28.1′ | 1366 |
| RG005 | 35°24.5′ | 82°57.8′ | 1520 | RG300 | 35°43.5′ | 83°13.0′ | 1558 | RRG13 | 35°3.75′ | 83°27.4′ | 961 |
| RG008 | 35°22.9′ | 82°58.4′ | 1737 | RG301 | 35°42.3′ | 83°15.3′ | 2003 | RRG41 | 35°3.32′ | 83°25.7′ | 776 |
| RG010 | 35°27.3′ | 82°56.8′ | 1478 | RG302 | 35°43.2′ | 83°14.8′ | 1860 | RRG12 | 35°2.84′ | 83°27.5′ | 1001 |
| RG100 | 35°35.1′ | 83°04.3′ | 1495 | RG303 | 35°45.7′ | 83°09.7′ | 1490 | RRG55 | 35°2.39′ | 83°27.3′ | 1035 |
| RG101 | 35°34.5′ | 83°05.2′ | 1520 | RG304 | 35°40.2′ | 83°10.9′ | 1820 | RRG96 | 35°2.72′ | 83°26.2′ | 894 |
| RG102 | 35°33.8′ | 83°06.2′ | 1635 | RG305 | 35°41.4′ | 83°07.9′ | 1630 | | | | |
| RG103 | 35°33.2′ | 83°07.0′ | 1688 | RG306 | 35°44.7′ | 83°10.2′ | 1536 | | | | |
| RG104 | 35°33.2′ | 83°05.2′ | 1587 | RG307 | 35°39.0′ | 83°11.9′ | 1624 | | | | |
| RG105 | 35°38.0′ | 83°02.4′ | 1345 | RG308 | 35°43.8′ | 83°10.9′ | 1471 | | | | |
| RG106 | 35°25.9′ | 83°01.7′ | 1210 | RG309 | 35°40.9′ | 83°09.0′ | 1604 | | | | |
| RG107 | 35°34.0′ | 82°54.4′ | 1359 | RG310 | 35°42.1′ | 83°07.3′ | 1756 | | | | |
| RG108 | 35°33.2′ | 82°59.3′ | 1277 | RG311 | 35°45.9′ | 83°08.4′ | 1036 | | | | |

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
