# Peer review of "An Expanded Investigation of Atmospheric Rivers in the Southern Appalachian Mountains and Their Connection to Landslides"

_atmosphere, doi:10.3390/atmos10020071_

Round 1
Reviewer 1 Report
Please, refer to the document attached.

Author Response
please find my comments in the attached document...

Reviewer 2 Report
Comments on “An Expanded Investigation of Atmospheric Rivers in 2 the Southern Appalachian Mountains and their 3 Connection to Landslides” by Miller et al.
The authors detected and characterized ARs in the southern Appalachian Mountains in two river basins, located 60 km apart, and examined the influence of ARs on extreme rainfall (ExtR), periods of elevated precipitation (ERTC) and landslide (LDSL) events over two time periods, the ‘recent’ (for both basins) and ‘distant’ (CRB only) past. Because extreme rainfall events in the southeastern US are complex in nature, the authors have done a great job in teasing out the effects of ARs in such events and their impacts. In particular, the authors took into account ARs, hurricanes among the others. Moreover, the authors have built relationships between ARs and landslides in the southeastern US. I believe that this study will add new information to the literature and is definitely worth publishing in this journal. I have some concerns elaborated below.
Line 50-61: The authors could consider including references reporting impacts of landfalling hurricanes on extreme precipitation in the US, in particular in the southeastern US.
Line 163-179: The method of detecting ARs in the southeastern US is more difficult than so do in the western US because they are mixed with frontal systems, tropical cyclone remnants and mesoscale convective systems. It is therefore necessary to add more discussions on the procedures by which the authors detect the ARs in study regions.
Line 201-206: Why did the authors choose to use the Twentieth Century Reanalysis (20CR) Project, rather than the NCEP/NCAR reanalysis data which are available for 1948 – present?
Line 225: can be initiated
Line 246: “all but three ARs are qualified”
Line 273: “Seasonal count of ARs impacting the”
Line 277-278: What do “{upper portion}” and “{lower portion}” mean in this context?
Line 290: The title of Figure 5 should be revised. 850 hPa geopotential height?
Line 378: “Elevated Rain Time Cluster (ERTC)” first appears in this sentence. What is the exact meaning of ERTC? The authors may add more explanations.
Line 332-334: The authors should focus on landfalling hurricanes instead of the occurrence of hurricanes because extreme rainfall in the study region is associated with the U.S. landfalling hurricanes. The authors may consider discussing “hurricane drought” as shown in Hart et al. (2016) and Klotzbach et al. (2018).
Line 348-350: “The most active years for landslide days in western North Carolina were 2003, 2005, 2009, and 2013, in ascending order, while few landslide days occurred over an extended period from 1997 through 2002 (Figures 8 and 9a).” The authors should add more discussions on Figure 8 in terms of the trends in LDSL and those in AR (Figure 8). It seems that after 2007, the year-to-year variation of LDSL and AR is in phase. Moreover, the authors can discuss the climate background in terms of El Nino/La Nina, North Atlantic Oscillation, Arctic Oscillation and the Pacific North American patterns during the years with extreme AR activity (Aryal et al. 2018)
Line 442-456: The analysis of weather patterns is quite intriguing. Figure 12 shows an intensification and westward march of the Bermuda high. This weather pattern is favorable for hurricane landfall (Murakami et al. 2016) and strengthened low-level jet stream.
Reference:
Aryal YN et al. (2018) Long term changes in flooding and heavy rainfall associated with North Atlantic tropical cyclones: Roles of the North Atlantic Oscillation and El Niño-Southern Oscillation. Journal of Hydrology 559:698-710.
Murakami, H et al. 2016: Statistical–Dynamical Seasonal Forecast of North Atlantic and U.S. Landfalling Tropical Cyclones Using the High-Resolution GFDL FLOR Coupled Model. Mon. Wea. Rev., 144, 2101–2123, https://doi.org/10.1175/MWR-D-15-0308.1
Hart, R.E., D.R. Chavas, and M.P. Guishard, 2016: The Arbitrary Definition of the Current Atlantic Major Hurricane Landfall Drought. Bull. Amer. Meteor. Soc., 97, 713–722,https://doi.org/10.1175/BAMS-D-15-00185.1
Klotzbach, P.J., S.G. Bowen, R. Pielke, and M. Bell, 2018: Continental U.S. Hurricane Landfall Frequency and Associated Damage: Observations and Future Risks. Bull. Amer. Meteor. Soc., 99, 1359–1376, https://doi.org/10.1175/BAMS-D-17-0184.1
Author Response

(The authors gave the same response as above.)

Reviewer 3 Report
This draft provides information related to impacts of AR or coincidence of AR and other atmospheric disturbances (tropical cyclones) on extreme rainfall and associated landslides in the southern Appalachian Mountains. The focus and results of this study are unique and interesting. Therefore, I am generally positive to recommend acceptance of this paper. However, presentation should be greatly improved before publication.
Specific comments
1. Generally quality of figure presentation is low. I cannot find information how did you make background maps in Figures 1 (righthand side) and 3 (how did you obtain this map? Any credit?). I cannot find scales of maps in Figures 1 (lefthand side) and 3a. Rainbow color schemes should not be used in Figures 5, 6, 12, 13, and 14. To improve visibility, colorbrewer2.org is useful tool for creating clear color scheme, for example (doi:10.1038/519291d).
2. I cannot accept Figure 9. This figure is not effective because it is hard to distinguish whether a bar in panel (a) corresponds to a bar in (b-c) or does not. Please add axis scale. Resolution of data plotted is 6-hr? day? month? To improve clarity, please highlight/add any marks if the lines are coincident, for example. Otherwise, this figure should be removed.
3. Vectors shown in Fig. 14 are not clear. They should be thicker and darker (how about black?). Shading color scheme can also be improve (blue shading can be removed, for example). Where is vector scale?
4. In figure 7, time series of running-mean value should be plotted (e.g. 11-yr running mean) instead of the values binned into decades.
5. Line 350-353: However, the two largest peaks (years 2009 and 2013) are coincident between LDSL and AR. This point should be mentioned here.
6. Line 39-40: Studies related to ARs mentioned here are only examined US, Europe and Antarctic ARs. Recently, importance of ARs over New Zealand (Kingston et al. 2016) and East Asia (Kamae et al. 2017) has also attracted much attention. Such works should also be included here.
7. Line 444: been -> be
References
Kingston D. G., D. A. Lavers, and D. M. Hannah, 2016: Floods in the Southern Alps of New Zealand: the importance of atmospheric rivers. Hydrol. Process., 30, 5063–5070.
Kamae, Y., W. Mei, and S.-P. Xie, 2017: Climatological relationship between warm season atmospheric rivers and heavy rainfall over East Asia. J. Meteor. Soc. Japan, 95, 411-431.
Author Response

(The authors gave the same response as above.)

Round 2
Reviewer 1 Report
Please, refer tho the attached document.

Author Response
Please find my response to your comments in the attached document.

Reviewer 3 Report
The authors greatly improve their manuscript (especially in terms of presentation) according to review comments. I don't have any further comments.
Author Response

(The authors gave the same response as above.)
